# Heterologous Machine Learning for the Identification of Antimicrobial Activity in Human-Targeted Drugs

**DOI:** 10.3390/molecules24071258

**Published:** 2019-03-31

**Authors:** Rodrigo A. Nava Lara, Longendri Aguilera-Mendoza, Carlos A. Brizuela, Antonio Peña, Gabriel Del Rio

**Affiliations:** 1Department of biochemistry and structural biology, Instituto de Fisiología Celular, UNAM, Mexico City 04510, Mexico; rnava@email.ifc.unam.mx; 2Computer Science Department, CICESE Research Center, Ensenada, Baja California 22860, Mexico; longendri@gmail.com (L.A.-M.); cbrizuel@cicese.mx (C.A.B.); 3Department of genetics, Instituto de Fisiología Celular, UNAM, Mexico City 04510, Mexico; apd@ifc.unam.mx

**Keywords:** machine-learning, antimicrobial peptide, non-peptidic antimicrobial compound, antimicrobial activity

## Abstract

The emergence of microbes resistant to common antibiotics represent a current treat to human health. It has been recently recognized that non-antibiotic labeled drugs may promote antibiotic-resistance mechanisms in the human microbiome by presenting a secondary antibiotic activity; hence, the development of computer-assisted procedures to identify antibiotic activity in human-targeted compounds may assist in preventing the emergence of resistant microbes. In this regard, it is worth noting that while most antibiotics used to treat human infectious diseases are non-peptidic compounds, most known antimicrobials nowadays are peptides, therefore all computer-based models aimed to predict antimicrobials either use small datasets of non-peptidic compounds rendering predictions with poor reliability or they predict antimicrobial peptides that are not currently used in humans. Here we report a machine-learning-based approach trained to identify gut antimicrobial compounds; a unique aspect of our model is the use of heterologous training sets, in which peptide and non-peptide antimicrobial compounds were used to increase the size of the training data set. Our results show that combining peptide and non-peptide antimicrobial compounds rendered the best classification of gut antimicrobial compounds. Furthermore, this classification model was tested on the latest human-approved drugs expecting to identify antibiotics with broad-spectrum activity and our results show that the model rendered predictions consistent with current knowledge about broad-spectrum antibiotics. Therefore, heterologous machine learning rendered an efficient computational approach to classify antimicrobial compounds.

## 1. Introduction

Drug-resistant microbes are one of the most important challenges for modern medicine [1] considering the increased rate in morbidity and mortality associated with antibiotic-resistant pathogens [2]. It is now commonly accepted that misuse of antibiotics is a major factor that promotes microbial resistance to these agents [3]; such is the case of broad-spectrum antibiotics that tend to promote resistance and are now prescribed in very restricted situations [4]. Furthermore, it has been noted that many non-antibiotic human-targeted drugs alter the gut microbiome in patients taking such drugs [5,6]. This alteration has been shown to be the consequence of a non-reported colateral antimicrobial activity, suggesting that microbe resistance to an antibiotic may emerge as a consequence of using those human-targeted drugs [7]. Furthermore, some antibiotics may have not been tested against the gut microbiome and may as well promote the emergence of resistant microbes. Since the experimental validation of antimicrobial activity for the gut microbiome requires tests on hundreds/thousands of cultivable and non-cultivable microorganisms and the number of new human-targeted drugs may include dozens of compounds, it is relevant to develop efficient computational strategies for the identification of secondary antimicrobial activity of human-targeted drugs. In the present work we present a computational strategy aimed to improve the identification of compounds with antimicrobial activity using machine-learning-based approaches.

Previous computational approaches to identify antibiotics using Quantitative Structure-Activity Relationships (QSAR) [8,9] and machine-learning-based [10,11] procedures have been reported. In these computational approaches, non-peptidic chemical compounds (from now on referred to as NPCC) are represented by chemical descriptors (e.g., *LogP*, molecular weight, polarizability) and each compound is labeled as antibiotic or non-antibiotic; then a clustering algorithm separates antibiotics from non-antibiotics. An important limitation of these previous studies is that the number of chemical compounds used to train the models is limited (less than one thousand NPCC have been described with antimicrobial activity) and the reliability of these models requires further improvement. Alternatively, antimicrobial peptides now accumulate in more than 10,000 in different databases [12,13,14], and several computational models have been reported to effectively classify antimicrobial from non-antimicrobial peptides [15,16,17]. Although peptides represent an important new focus to develop pharmaceuticals, most human-targeted drugs are NPCC; therefore computational models to identify antimicrobial activity in these compounds should focus on NPCC. The need to use common molecular descriptors between polypeptides and NPCC has been previously noted for protein-ligand recognition and protein folding, as a fundamental aspect to deal with induced-fit or conformer selection mechanisms for molecular recognition [18]; the aim of this work though, is not to find common descriptors to peptides and NPCC since there are already packages that solve this problem (see below). Here we propose that combining peptides and NPCC increases the training set size and this should improve the reliability of the computational models. The present work tests this proposal and validates the idea that heterelogous (NPCC and peptides) training sets render the best classifying models. We then show how this improved model may assist in the identification of broad-spectrum antibiotics on FDA-approved NPCC.

## 2. Results

### 2.1. Training and Testing Gut Antimicrobial Classifiers

Building data sets to combine peptides and NPCC required the use of molecular descriptors common to both types of compounds; in our case, we used 1444 descriptors calculated by PadelDescriptor (see Methods). Then, to identify the best machine-learning model to classify gut antimicrobials, three groups of training sets were used (see Table 1). The first group included only peptides (TrOnlyPeptides), the second group comprises 4 sets and included only NPCC (TrNPCC1-4) and the third group combined these two previous sets (TrHeterologous1-4) resulting in a total of 9 training sets (see Table 1); this rendered a total of 45 training sets. These 45 sets were further processed to substitute any null or "Infinity" values using three different approaches, and a reduction of dimensions was performed via principal-component analysis (PCA, see Methods). This procedure rendered a total of 50 Training Sets; all these sets are included in Appendix A.

Nine testing sets were built using the NPCC recently reported by Maier et al. [7] with and without gut antimicrobial activity (see Table 2). The same processing of these testing sets was performed as in the case of the training sets (see above), rendering again a total of 50 data sets (see Appendix A). Please note that in both training and testing sets all peptides included were tested against only one gut microbe assayed against the NPCC used in these sets and that although there are many more peptides than NPCC in our training and testing sets, this imbalance is not relevant to find the border between antimicrobials and non-antimicrobials compounds.

Five different statistical parameters (adjusted estimated error rate on the training set (AEER); correctly classified instances in the training set after splitting 33% for testing (%Split); 10-fold cross-validation (%10FCV); correctly classified instances on the testing set (%CC); area under the receiver operator characteristic curve on the testing set (AUROC)) that evaluated the performance on either the training or testing sets (see Methods) were used to identify the best classifier.

As shown in Figure 1, the best models included heterologous compounds (peptides and NPCC): circles in Figure 1 represent heterologous training sets and accumulate on the upper part of Figure 1, that is, those models with highest statistical parameters evaluating the model performance (the actual data in this figure for these models are included in Appendix A). Treating the training set rendering the best model with the K-nearest neighbor or mean-imputation approaches did not improve the performance of the best model (see Appendix A and its corresponding test set in Appendix A; Appendix A and their corresponding test sets in Appendix A).

Yet, none of these models surpassed the others in all 5 parameters. To aid in the visualization of this aspect of our results, Figure 1 displays the values in descending order from left to right; therefore, the models on the left side of the plot have better scores than those on the right. For instance, models using heterologous (represented by circles) testing sets (the red and blue circles, corresponding with the statistical parameters correctly classified instances and *AUROC*, respectively) laying on the left side of Figure 1, have better performance than those models using heterologous testing sets on the right side of the plot, yet, those on the right side including either heterologous or non-heterologous training sets (green and yellow circles or triangles) have better scores than those models using heterologous or non-heterologous training sets on the left side of the plot. The models in the middle of the plot have on the other hand, intermediate performances. Please note that the statistical parameter adjusted estimated error rate is the value that AutoWeka optimizes, hence for all the reported models is close to 1.0 and consequently does not contribute to differentiate the performance of the models. This statistical parameter is shown in Figure 1 to note that all models have similar error rates, yet different statistical parameters, hence, the best model obtained from AutoWeka cannot be selected simply by considering the error rate value reported.

Thus, to aid in the identification of the best models, we used a previous score developed by our group that takes into account multiple statistical parameters, the Combined Score or simply *CScore* [19]:(1)$$CS{core}_{i}=\frac{1}{5}\sum_{n=1}^{5}\left[\sqrt{\frac{MaxS_{n}-S_{i,n}}{MaxS_{n}-MinS_{n}}}\right]$$
where *MaxS_n_* and *MinS_n_* represent the maximum and minimum scores for a given statistical parameter *n* over all models; *S_i,n_* is the score observed for a given statistical parameter *n* and model *i*; *n* represents the index of the statistical parameter to evaluate (in our case were 5 parameters: AEER, %Split, %10FCV, %CC and AUROC). Thus, formula 1 calculates *CScore* for each model *i*.

*CScore* averages the difference of each statistical parameter to its best value (e.g., true-positive rate best value is 1, so the difference between the observed true positive rate and 1 is included in the *CScore*), therefore the lower the *CScore* value the better the classifying model. Figure 2 (and Appendix A) shows that the five best models are those using heterologous training sets (the ones below the 0.3 line in Figure 2). Furthermore, we noticed that the top 5 best models overlapped on average in more than 70% of their classifications hence, these were mainly redundant (see Figure 3).

Therefore, we selected the best model based on the lowest *CScore*; such model was built using the RandomCommittee algorithm (see Appendix A for the algorithm parameters) on the TrHeterologous1 set (see Table 1) that included 86 molecular PCA-reduced descriptors and achieved the following performance: AEER: 0.99955; %Split: 87.4; %10FCV: 87.3; %CC: 77.2; AUROC: 0.83 (model named TrHeterologous1-Reduced-With-99M-CRC20_CRF in Appendix A; the corresponding data set for this model is reported in Appendix A).

### 2.2. Identifying Broad-Spectrum Antibiotics among FDA-Approved Compounds

We used the best model to predict NPCC with expected gut antimicrobial activity among FDA-approved drugs. The motivation to perform this prediction is not for testing purposes, as in the case of the training and testing sets used before. Hence, the set of compounds used in this prediction stage is referred to as the discovery set, because we aimed to discover potential compounds with gut antimicrobial activity. We used 756 FDA-approved compounds included in the ZINC database (see Methods) that were not part of the training or testing sets; these compounds included 111 antimicrobials and 645 compounds without any known antimicrobial activity; we also added 73 NPCC that included 22 antifungal compounds and 51 without any reported antifungal activity (see Appendix A). We have previously reported that these 22 antifungals work through a mechanism (alter calcium intake [20]) different from antibacterial compounds (e.g., penicillin derivates, sulphonamides, etc), thus we expected our model to predict few of these compounds as antibacterials. FDA-approved compounds on the other hand are expected not to have, or to have minor, gut antimicrobial activity otherwise their secondary gastrointestinal effects would be significant. We would expect that FDA-approved drugs would be less likely predicted to act against non-athogenic gut microbes than antifungals. To evaluate the reliability of our predictions using the discovery set, we considered that antibiotic compounds against the non-pathogenic gut flora among the FDA-approved drugs should be considered broad-spectrum antibiotics; please note that our classifier was not trained to predict this class of antibiotics, yet the combination of the predictions of our classifier on the FDA-approved drugs would render this information. The definition of broad-spectrum antibiotics is somehow arbitrary, for instance, it is considered that antibiotics that act on G(+) and G(−) are broad-spectrum antibiotics for some authors, while those acting against pathogenic and non-pathogenic microorganisms are classified as broad-spectrum antibiotics by others [21,22]. The list of broad-spectrum antibiotics was obtained from five recent works (see Methods), including 19 broad-spectrum and 3 narrow-spectrum antibiotics (see Appendix A). We were able to identify 72 true positives (FDA-approved antibiotics against pathogenic microbes predicted to act against non-pathogenic gut microbes) in the discovery set that we predicted should be considered as broad-spectrum antibiotics (see Table 3).

The actual data for this table can be found in Appendix A.

From these 72 antimicrobials, only 16 had been annotated as broad-spectrum antibiotics and 3 as narrow-spectrum antibiotics (see Appendix A). Hence, we propose that these 3 annotated narrow-spectrum antibiotics should be considered more likely as broad-spectrum antibiotics (see Table 4).

On the other hand, among the 61 false negatives, 3 compounds were annotated as broad-spectrum antibiotics (see Appendix A). This annotation is consistent with our predictions, since these antibiotics directed towards pathogenic microorganisms are unlikely to affect the non-pathogenic gut microbes. Furthermore, 17 out of the 22 antifungal compounds were predicted as antimicrobials.

Thus, in total we were able to correctly identify 16 out of the 19 known broad-spectrum antibiotics and we suggest that 3 of the annotated narrow-spectrum antibiotics should be re-evaluated; hence, the reliability to identify broad-spectrum antibiotics was 84.2%. Furthermore, our results suggest that 56 (61 true negatives less 5 antifungals) (50.4%) out of 111 antibiotics approved by the FDA included in our discovery set are unlikely to affect gut microbes. In comparison, 5 (22.7%) out of 22 antifungals were predicted not to act against the gut microbes (see Appendix A). Thus, it is twice as much less likely that FDA-approved antibiotics would be toxic against gut microbes than antifungals.

## 3. Discussion

The identification of antimicrobial compounds assisted by machine-learning techniques has multiple advantages, such as reduction of the invested time to develop novel pharmaceuticals or to flag molecules that could have secondary antimicrobial activity [17]. An important aspect of these techniques is how to improve the reliability of these predictions. One way to achieve this is to increase the number of examples in the training and testing sets. In this work we propose that it is possible to use chemical compounds of different nature (peptides and NPCC) that are commonly modeled separately as antimicrobials to improve the reliability of the predictions. Here we show that indeed, the training sets that rendered the best classifiers of antimicrobial compounds were heterologous, those including NPCC and peptides (see Figure 1 and Figure 2). We can compare our best classifier with previous works in terms of the learnability of our classes, that is, how well gut antimicrobial compounds are differentiated from non-antimicrobial gut compounds. In that sense, the numeric performance achieved by the best classifier on the testing set (AUC = 0.83) is comparable with the performance achieved with one of the best antimicrobial peptide classifiers (AUC = 0.85) recently reported [23], indicating that the learnability of heterologous training sets is as good as those of only peptides.

Another important aspect of our work is the molecular descriptors obtained to best classify gut antimicrobial compounds that included both peptides and NPCC. Although our goal was not to identify common descriptors for NPCC and peptides (these are already calculated by available packages, see Methods), we did look for those descriptors that are relevant to learn the difference between antimicrobials from non-antimicrobials. Our results indicate that the solution to this problem requires the transformation of 86 computed molecular descriptors, suggesting that other molecular descriptors, most likely associated to these 86 descriptors, may improve the current best-model performance.

In terms of improving the performance reported in this work, it is worth mentioning that we used peptides that were not tested by Maier et al. [7] yet, these peptides had reported antibiotic activity against at least one microorganism (*Escherichia coli*) found in the gut and tested by Maier and collaborators. On the other hand, the NPCC included in our work had antibiotic activity against at least one of the 40 gut microorganisms tested by Maier and collaborators. Hence, one alternative approach to improve the performance of classifiers aimed at identifying gut microorganisms would be to include antibiotics that target more common gut microorganisms; that would require further experimental data that is not currently available at present.

To the best of our knowledge, no previous machine-learning efforts to assist in the identification of broad-spectrum antibiotics have been reported; here the definition of broad-spectrum antibiotics was restricted to those acting against both pathogenic and non-pathogenic microorganisms. Hence, using a classifier trained to identify gut non-pathogenic antimicrobial compounds to predict this activity in FDA-approved antibiotics targeted against pathogenic microorganisms represents a way to identify broad-spectrum antibiotics. Our results suggest that half of the FDA-approved antibiotics are likely to have antimicrobial activity against the gut microorganisms indicating that these require further testing or investigation. For instance, two annotated narrow-spectrum antibiotics, amoxicillin and cephalexin, that were predicted to alter gut microbes are known to affect the gastrointestinal flora [24]. On the other hand, the broad-spectrum antibiotic ceftaroline fosamil recently approved by the FDA to treat bacterial pneumonia and skin infections, which was not predicted to affect the gut flora, was reported to have minor gastrointestinal effects during clinical trials [25].

How significant is our finding that almost half of the FDA-approved antibiotics are predicted to have a broad-spectrum activity? To address this question, we included in the discovery set a group of antifungal compounds. All microorganisms used to train our models were bacteria, hence we expected that these antifungals that act through a mechanism different from those reported for bacteria would be unlikely predicted to act against bacteria; lets refer to this negative prediction as *expectation-antifungal*. On the other hand, most FDA-approved antibiotics should unlikely present antibiotic activity against gut microbes, otherwise these would frequently have secondary gastrointestinal effects on patients; lets refer to this negative prediction as *expectation-FDA*. Then, to address the significance of our findings about broad-spectrum antibiotics requires evaluating *expectation-antifungal* and *expectation-FDA*; if FDA-approved drugs are less likely to act on gut microbes than antifungals then *expectation-FDA* < *expectation-antifungal*. Indeed, we observed that FDA-approved antibiotics are twice as much less likely to act against gut microbes than antifungals. Thus, our results indicate that even when FDA-approved antibiotics are safer (do not act against non-pathogenic resident gut bacteria) than our control group (antifungals), we identified some of these compounds that need to be re-assessed as potential promoters of resistance among microbes for their potential broad-spectrum activity.

In summary, we report a computational approach to use heterologous antimicrobial compounds (peptides and non-peptides) to improve the discriminatory power of machine-learning approaches. We show that training a classifier to identify antibiotics against the gut flora using heterologous training sets correctly anticipate adverse gastrointestinal reactions in patients receiving these antibiotics.

## 4. Materials and Methods

### 4.1. Materials

Peptides included in the training sets were obtained from the non-redundant data set of 20 public databases (see Table 5). Testing sets were derived from the work reported by Maier and collaborators (see Appendix A). Finally, a discovery set containing 750 FDA-approved drugs for treating human infectious diseases and 76 antifungal drugs was built from the ZINC database [26]. Molecular descriptors were computed with PadelDescriptor [27]. For every training and test set, we performed five different approaches to process the molecular descriptors for each peptide and/or NPCC. These included: no processing; eliminate every null value; substitute every “Infinity” value for 0 or 99,999,999; reduction of the dimensionality applying a principal component analysis implemented in WEKA package (see below). Since the substitution of Infinity values for 0 or 99,999,999 is not a conventional strategy, we performed an imputation of the Infinity and null values using the K nearest neighbor or mean imputation approaches, but only on the best model data set for comparison. That is, from the 9 training sets we generated a total of 45 training sets following the different approaches described before; the same applies to the 9 testing sets. For the discovery set only the transformation applied to the best classifier was performed.

### 4.2. Method

To identify the best model to classify gut antimicrobial compounds, we followed a systematic method previously reported by our group [46]. Briefly, given the training sets, 52 different machine-learning algorithms implemented in WEKA [47] and their parameters were systematically analyzed to identify the algorithm, parameters and molecular descriptors that renders the lowest possible error in classification; this systematic analysis was performed by the Bayesian optimization algorithm implemented in AutoWEKA [48]. We ran AutoWEKA against any training set for 10, 90, 720, 2880 and 4320 minutes to identify when the optimization has reached a plateau in the classification error. Afterwards, a 10-fold cross validation and 67% split tests were performed in WEKA. Finally, these classifiers were evaluated against their corresponding testing sets. Two statistical parameters were chosen to evaluate the performance of the classifiers during the testing, including: Area under the ROC curve and correctly classified instances on the testing set. Therefore, a total of 5 statistical parameters were used to define the best classifiers, three for the training phase (adjusted estimated error rate on the training set; correctly classified instances in the training set after splitting 33% for testing; 10-fold cross-validation) and two for the testing phase (AUROC and correctly classified instances).

To identify the intersection set between the top 5 classifiers, we compared the predictions of these classifiers rendering 10 possible pairs of predictions on the discovery set; we used this set because not every classifier had the same testing set. The best model was identified using a combined score (see formula 1): the model with the lowest combined score was chosen. The model then was used to predict gut antimicrobial compounds in the discovery set using WEKA command line (see Appendix A). To annotate as broad-spectrum or narrow-spectrum antibiotics, we used five different previous works that classified antibiotic action [22,49,50,51,52].

## Figures and Tables

**Figure 1 molecules-24-01258-f001:**
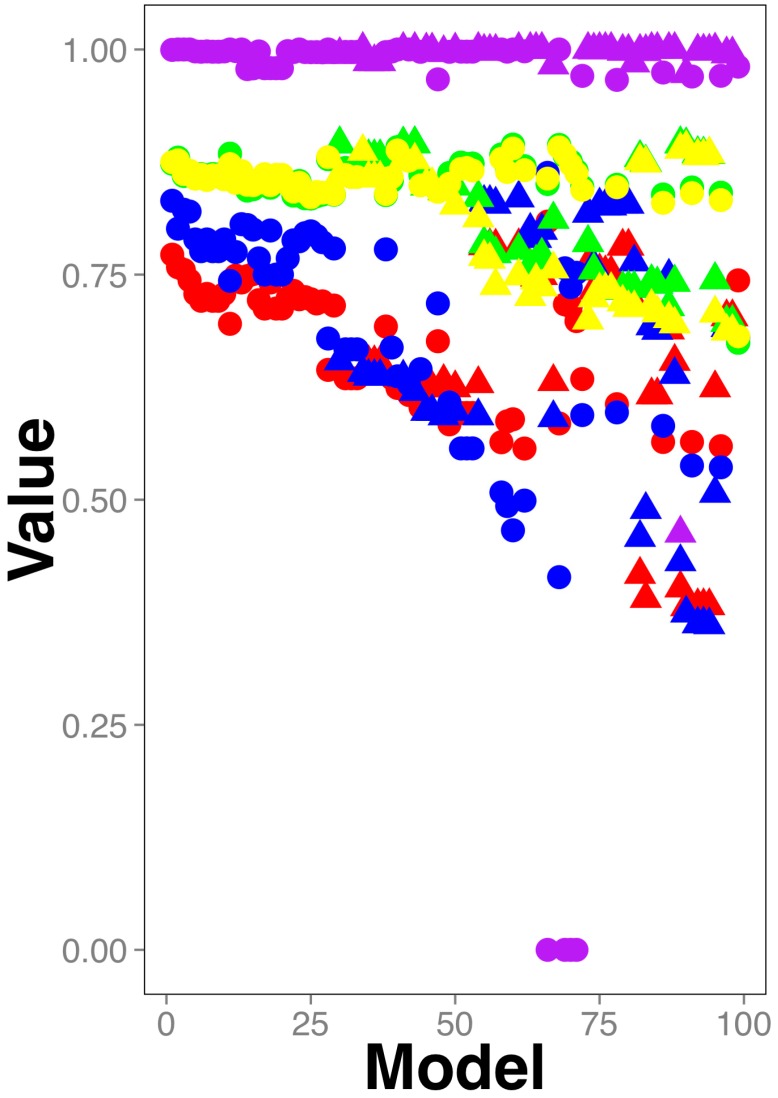
Classifiers performance. Five statistical parameters (yellow circle: Correctly classified instances in the test set after 67% of split validation of the training set; green circle: 10-fold cross-validation on training set; red circle: Correctly classified instances on the testing set; blue circle: AUROC on the testing set; purple circle: adjusted estimated error rate on the training set where the minimum error was represented by number 1.0) are sorted from highest to lowest values. Hence, the best parameter values are located on the left-upper part of the figure. The models using heterologous data are represented as circles; triangles are used otherwise. The actual data of this plot can be found in Appendix A.

**Figure 2 molecules-24-01258-f002:**
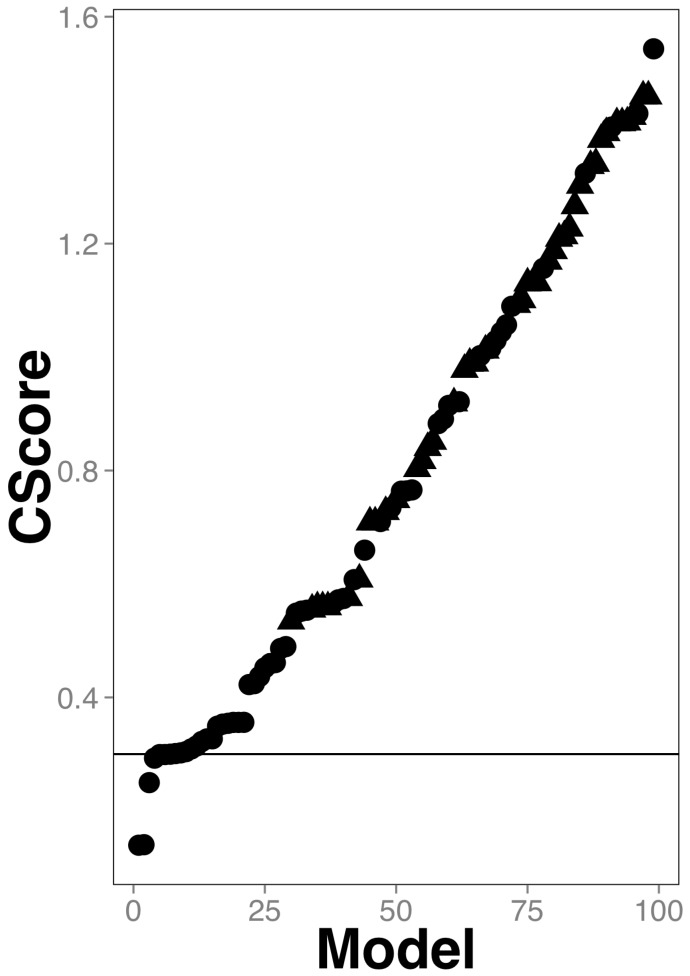
Classifiers combined scores. A circle represents each model; the best model has the lowest *CScore*. The line represents the *CScore* = 0.3, that separates the top 5 models from the rest. The models using heterologous data are represented as circles; triangles are used otherwise. The actual data of this plot can be found in Appendix A.

**Figure 3 molecules-24-01258-f003:**
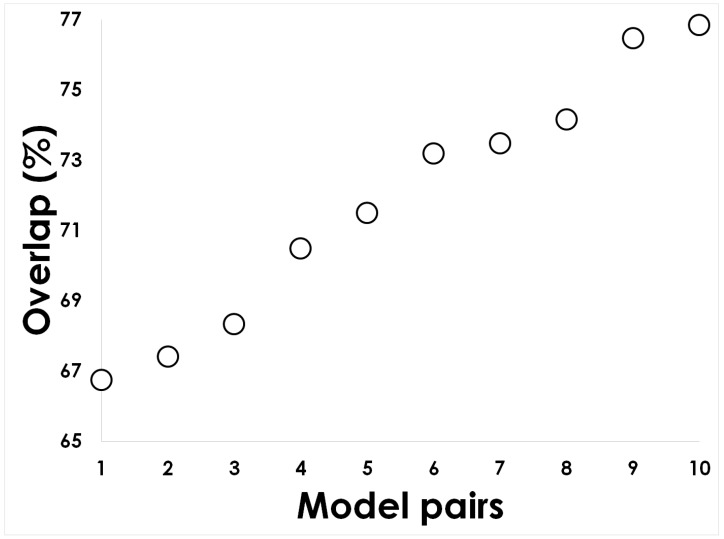
Classifiers overlap. The predictions of antimicrobial compounds of the top 5 models were compared to quantify their overlap. The image shows the 10 pairs of models generated from these 5 top models. The comparison was performed on the discovery set (see Methods) because not every model had the same testing set.

**Table 1 molecules-24-01258-t001:** Training data sets.

Training Set	Entries	Description
TrOnlyPeptides	11,546	8000 antimicrobial peptides, 3546 peptides with no known antimicrobial activity
TrNPCC1	431	164 antimicrobial non-peptides, 267 non-peptides with no known antimicrobial activity
TrNPCC2	430	164 antimicrobial non-peptides, 266 non-peptides with no known antimicrobial activity
TrNPCC3	430	164 antimicrobial non-peptides, 266 non-peptides with no known antimicrobial activity
TrNPCC4	431	164 antimicrobial non-peptides, 267 non-peptides with no known antimicrobial activity
TrHeterologous1	6204	4164 antimicrobial compounds (4000 peptides and 164 non-peptidic compounds), 2040 no antimicrobial compounds (1773 peptides and 267 non-peptidic compounds)
TrHeterologous2	6203	4164 antimicrobial compounds (4000 peptides and 164 non-peptidic compounds), 2039 no antimicrobial compounds (1773 peptides and 266 non-peptidic compounds)
TrHeterologous3	6203	4164 antimicrobial compounds (4000 peptides and 164 non-peptidic compounds), 2039 no antimicrobial compounds (1773 peptides and 266 non-peptidic compounds)
TrHeterologous4	6204	4164 antimicrobial compounds (4000 peptides and 164 non-peptidic compounds), 2040 no antimicrobial compounds (1773 peptides and 267 non-peptidic compounds)

The original NPCC from Maier et al. [7], here referred to as OnlyNonPeptides, was used to build TrNPCC1 by taking only the odd listed compounds, TrNPCC2 by taking even listed compounds, TrNPCC3 and TrNPCC4, included the first and second half of the data set respectively. The OnlyPeptides data set was divided to generate TrHeterologous1, TrHeterologous2, TrHeterologous3 and TrHeterologous4 by taking the odds listed peptides, even listed peptides, first and second half, respectively. Then, these TrHeterologous1-4 data sets with peptides were combined with the TrNPCC1-4 to complete these sets.

**Table 2 molecules-24-01258-t002:** Testing data sets.

Testing Set	Entries	Description
TeOnlyPeptides	861	328 antimicrobial and 533 non-antimicrobial non-peptides
TeNPCC1	430	164 antimicrobial non-peptides, 266 non-peptides with no known antimicrobial activity. Same as TrNPCC2.
TeNPCC2	431	164 antimicrobial non-peptides, 267 non-peptides with no known antimicrobial activity. Same as TrNPCC1.
TeNPCC3	431	164 antimicrobial non-peptides, 267 non-peptides with no known antimicrobial activity. Same as TrNPCC4.
TeNPCC4	430	164 antimicrobial non-peptides, 266 non-peptides with no known antimicrobial activity. Same as TrNPCC3.
TeHeterologous1	430	Same as TeNPCC1.
TeHeterologous2	431	Same as TeNPCC2.
TeHeterologous3	431	Same as TeNPCC3.
TeHeterologous4	430	Same as TeNPCC4.

The original NPCC from Maier et al. [7], here referred to as OnlyNonPeptides, was used to build all Testing Sets. TeOnlyPeptides was built taking all the 861 listed compounds. TeNPCC1 and TeHeterologous1 were built by taking only the even listed compounds. TeNPCC2 and TeHeterologous2 included only the odd listed compounds. TeNPCC3 and TeHeterologous3 included the second half of OnlyNonPeptides, TeNPCC4 and TeHeterologous4 included the first half of the data set. Testing sets were built so they were the complement of the compounds listed for their Training sets, so, for example, if a training set was built using the even listed compounds (e.g., TrNPCC1), its Testing set would be built with the odd listed compounds (e.g., TeNPCC1). Heterologous Testing Sets were the same as OnlyNonPeptides Testing sets, due to the fact that the interest compounds are of non-peptidic nature.

**Table 3 molecules-24-01258-t003:** Confusion matrix for the discovery set.

	Predicted Gut Antimicrobial	Predicted No Antimicrobial
Pathogenic antimicrobial	72	61
No antimicrobial	140	556

**Table 4 molecules-24-01258-t004:** True pathogenic antimicrobials predicted by the best classifier on the discovery set.

Compound Name	Annotation
Amoxicillin	Narrow spectrum
Phenoxymethylpenicillin	Narrow spectrum
Cephalexin	Narrow spectrum

**Table 5 molecules-24-01258-t005:** Antimicrobial peptide databases used in the present study.

Database	Focused on	Reference
BACTIBASE	Bacteriocins	[28]
Bagel	Bacteriocins	[29]
CAMP	General and Patented AMPs	[14]
DADP	Anuran AMPs	[30]
DAMPD	General AMPs *	[31]
DBAASP	General AMPs	[13]
Defensins	Defensins	[32]
HIPdb	Anti-HIV peptides	[33]
LAMP	General and Patented AMPs	[34]
MilkAMP	AMPs of dairy origin	[35]
PhytAMP	Plant AMPs	[36]
PenBase	Penaeidin AMPs	[37]
Peptaibol	Peptaibols	[38]
RAPD	Recombinant AMPs	[39]
AMPer	Eukaryotic AMPs	[40]
UniprotKb	General AMPs	[41]
YADAMP	General AMPs	[42]
AMSDb	Eukaryotic AMPs	[43]
APD	General AMPs	[44]
AVPdb	Antiviral peptides	[45]

* AMPs stands for Antimicrobial Peptides.

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
