# Peer review of "Heterologous Machine Learning for the Identification of Antimicrobial Activity in Human-Targeted Drugs"

_molecules, 2019, doi:10.3390/molecules24071258_

Round 1

Reviewer 1 Report

Authors presented a machine-learning-based approach to identify antibiotic compounds against the non-pathogenic gut flora by employing a peptide and non-peptide heterologous training dataset. I found this idea well-founded, even though authors avoided a physicochemical justification of this choice (for instance, in Phys. Chem. Chem. Phys., 2018, 20, 17148 authors can find a related approach in which peptide datasets are used to construct a unified scoring function for the evaluation of protein and non-protein interactions). Employing the appropriate molecular descriptors, machine-learning techniques can take advantage of the huge number of peptide datasets for improving the reliability of their computational predictions. Nevertheless, I found important issues in the analysis methodology that prevent them obtaining the subsequent conclusions. Thus, I would be happy to read again the manuscript after all the points described below have been addressed.

1. The heterologous training sets employed by authors are composed by ~6200 antimicrobial compounds, in which 93% are peptides. First, I wonder the reason of this size and the peptide/non-peptide proportion, and why these variables have not been optimized previously. What is the role of the 7% of non-peptide compounds in the set?

2. The description of the Fig. 1 in the Results section does not match with the information showed in Fig. 1. For instance, in the line 117 authors claim that:

"As shown in Figure 1, the top 5 best models included heterologous compounds (circles accumulate on the left of Figure 1): peptides and NPCC."

It is impossible to know which are the 5 best models using only Fig. 1.

In the line 130 it is written:

"Figure 1 shows that the models on the left side of the plot have better scores represented by the red and blue circles (statistical parameters for testing sets; see Figure 1) than those on the right side of the plot, yet, those on the right side have better scores represented by the green and yellow circles (statistical parameters for the training sets; see Figure 1)."

I found really confusing this sentence taking into account that some scores obtained with non-peptidic sets (triangles) have similar values as the best heterologous sets for the "blue" and "red" statistical parameters.

Moreover, I wonder the role of the statistical parameter violet: adjusted estimated error rate on the training set where the minimum error was represented by number 1.0, which is clear from the data showed in Fig. 1 that it barely discern among models.

3. In the caption of Fig.1 authors clarify that scores "are sorted from highest to lowest values", but scores in the figure does not follow this sorting. I guess all these scores have been sorted according to the CScore values obtained for each model, although the CScore is defined afterwards. I would like to ask the authors to revise this issue.

4. Authors explained that they searched for the molecular descriptors common to peptides and non-peptides among 1444 candidates. Subsequently, 86 molecular descriptors were selected by PCA. I would like to ask to the authors to include in the Supplementary Section the classification of these 86 molecular descriptors.

5. I wonder why the authors did not compare their predictions of the anti-microbial activity of the discovery set by employing the best heterologous training set and the best non-peptidic training set. Without this comparison, it is difficult to conclude the improvement achieved by using big-size heterologous training sets, which I think it is one of the objectives of this work.

Author Response

1.     Authors presented a machine-learning-based approach to identify antibiotic compounds against the nonpathogenic gut flora by employing a peptide and non-peptide heterologous training dataset. I found this idea well-founded, even though authors avoided a physicochemical justification of this choice (for instance, in Phys. Chem. Chem. Phys., 2018, 20, 17148 authors can find a related approach in which peptide datasets are used to construct a unified scoring function for the evaluation of protein and non-protein interactions).

We appreciate the reference to this work; we were not aware of it. We are now including a reference to it in the Introduction “The need to use common molecular descriptors between polypeptides and NPCC has been previously noted for protein-ligand recognition and protein folding, as a fundamental aspect to deal with induced-fit or conformer selection mechanisms for molecular recognition [19]; the aim of this work though, is not to find common descriptors to peptides and NPCC since there are already packages that solve this problem (see below). Here we propose that combining peptides and NPCC increases the training set size and this should improve the reliability of the computational models”. We also include some discussion about our findings and those reported in the aforementioned reference: “Another important aspect of our work is the molecular descriptors obtained to best classify gut antimicrobial compounds that included both peptides and NPCC. Although our goal was not to identify common descriptors for NPCC and peptides (these are already calculated by available packages, see Methods), we did look for those descriptors that are relevant to learn the difference between antimicrobials from non-antimicrobials. Our results indicate that the solution to this problem requires the transformation of 86 computed molecular descriptors, suggesting that other molecular descriptors, most likely associated to these 86 descriptors, may improve the current best-model performance”.

2.     The heterologous training sets employed by authors are composed by ~6200 antimicrobial compounds, in which 93% are peptides. First, I wonder the reason of this size and the peptide/non-peptide proportion, and why these variables have not been optimized previously. What is the role of the 7% of non-peptide compounds in the set?

If we understand correctly, the main concern of this reviewer is the well-known effect in the reliability of machine-learning approaches on imbalanced training sets; in that sense, our training and testing sets tried to balance the number of antimicrobial and non-antimicrobial compounds, which are the classes we wanted to learn. That is, there is a small imbalance for the machine-learning algorithm to learn the difference between antimicrobial and non-antimicrobial compounds. Our goal was not to learn the differences between peptides and NPCC, hence we did not try to optimize for this imbalance.

Thus, the role of the 7% of NPCC in the set was to include a region in the multidimension space used to represent peptides and NPCC that can be learned to differentiate antimicrobials from non-antimicrobials; more importantly, that 7% was moving the borders of antimicrobials to those acting against gut microbes. The fact that we observed a clear improvement in antimicrobial classification in comparison with training sets composed exclusively of peptides or NPCC supports the relevance to include this set of NPCC in our training sets.

To clarify this, we now state in the results section: “Please note that in both training and testing sets all peptides included were tested against only one gut microbe assayed against the NPCC used in these sets and that although there are many more peptides than NPCC in our training and testing sets, this imbalance is not relevant to find the border between antimicrobials and non-antimicrobials compounds”.

3.     The description of the Fig. 1 in the Results section does not match with the information showed in Fig. 1. For instance, in the line 117 authors claim that:

"As shown in Figure 1, the top 5 best models included heterologous compounds (circles accumulate on the left of Figure 1): peptides and NPCC."

It is impossible to know which are the 5 best models using only Fig. 1.

Thanks for this note. To clarify what we mean about figure 1, we now state in the results section: “As shown in Figure 1, the best models included heterologous compounds (peptides and NPCC): circles in figure 1 represent heterologous training sets and accumulate on the upper part of Figure 1, that is, those models with highest statistical parameters evaluating the model performance (the algorithm, parameters and training sets used for these models are included in supplemental Table S19)”.

    In the line 130 it is written:

    "Figure 1 shows that the models on the left side of the plot have better scores represented by the     red and blue circles (statistical parameters for testing sets; see Figure 1) than those on the right     side of the plot, yet, those on the right side have better scores represented by the green and         yellow circles (statistical parameters for the training sets; see Figure 1)."

    I found really confusing this sentence taking into account that some scores obtained with         nonpeptidic sets (triangles) have similar values as the best heterologous sets for the "blue" and     "red" statistical parameters.

Thanks for the note. To clarify this, we now state that in the results section: “Yet, none of these models surpassed the others in all 5 parameters. To aid in the visualization of this aspect of our results, Figure 1 displays the values in descending order from left to right; therefore, the models on the left side of the plot have better scores than those on the right. For instance, models using heterologous (represented by circles) testing sets (the red and blue circles, corresponding with the statistical parameters correctly classified instances and AUROC, respectively) laying on the left side of Figure 1, have better performance than those models using heterologous testing sets on the right side of the plot, yet, those on the right side including either heterologous or non-heterologous training sets (green and yellow circles or triangles) have better scores than those models using heterologous or non-heterologous training sets on the left side of the plot. The models in the middle of the plot have on the other hand, intermediate performances”.

    Moreover, I wonder the role of the statistical parameter violet: adjusted estimated error rate on     the training set where the minimum error was represented by number 1.0, which is clear from the     data showed in Fig. 1 that it barely discern among models.

This observation is correct and we realized that we omitted the motivation to show this statistical parameter in Figure 1. To clarify this, we now state in the results section: “Please note that the statistical parameter adjusted estimated error rate is the value that AutoWeka optimizes, hence for all the reported models is close to 1.0 and consequently does not contribute to differentiate the performance of the models. This statistical parameter is shown in Figure 1 to note that all models have similar error rates, yet different statistical parameters, hence, the best model obtained from AutoWeka cannot be selected simply by considering the error rate value reported”.

4.     In the caption of Fig.1 authors clarify that scores "are sorted from highest to lowest values", but scores in the figure does not follow this sorting. I guess all these scores have been sorted according to the CScore values obtained for each model, although the CScore is defined afterwards. I would like to ask the authors to revise this issue.

Figure 1 is indeed sorted from highest to lowest; this can be appreciated by noticing the trend towards lower values from left to right of symbols colored in red, blue, yellow and green. We hope that with the changes in the description of Figure 1 noted above, this point is clarified.

5.     Authors explained that they searched for the molecular descriptors common to peptides and non-peptides among 1444 candidates. Subsequently, 86 molecular descriptors were selected by PCA. I would like to ask to the authors to include in the Supplementary Section the classification of these 86 molecular descriptors.

We appreciate the note; we recognized that this was not explicitly stated in the description of our work, even though the requested data was already part of the supplementary information. Yet, we realized that some information was missing, that is, we only described the name of the algorithm and data set used to train this algorithm, but we did not include the parameters obtained for the best model nor for the other models; this information now is in supplemental table S21. We have also changed the description of the training and testing sets and separated every set in individual files as supplemental material. These changes are reflected in the results section: “These 45 sets were further processed to substitute any null or "Infinity" values using three different approaches, and a reduction of dimensions was performed via principal-component analysis (PCA, see Methods). This procedure rendered a total of 50 Training Sets; all these sets are included in supplemental Tables S1A,B,C,D,E-S9A,B,C,D,E.”; “The same processing of these testing sets was performed as in the case of the training sets (see above), rendering again a total of 50 data sets (see supplemental Tables S10A,B,C,DE-S18A,B,C,D,E).”; “Treating the training set rendering the best model with the K-nearest neighbor or mean-imputation approaches did not improve the performance of the best model (see supplemental Tables S6G, S6I, S6J and its corresponding test set in Table S15G; supplemental Tables S6K, S6L and their corresponding test sets in supplemental Tables S15I and S15J)”. Finally, we also specify in the results section the missing information about the algorithm parameters: “Therefore, we selected the best model based on the lowest CScore; such model was built using the RandomCommittee algorithm (see supplemental Table S21 for the algorithm parameters) on the TrHeterologous1 set (see Table 1) that included 86 molecular PCA-reduced descriptors and achieved the following performance: AEER: 0.99955; %Split: 87.4; %10FCV: 87.3; %CC: 77.2; AUROC: 0.83 (model named TrHeterologous1-Reduced-With-99M-CRC20_CRF in supplemental Table S21; the corresponding data set for this model is reported in supplemental Table S6E)”.

6.     I wonder why the authors did not compare their predictions of the antimicrobial activity of the discovery set by employing the best heterologous training set and the best non-peptidic training set. Without this comparison, it is difficult to conclude the improvement achieved by using big-size heterologous training sets, which I think it is one of the objectives of this work.

The purpose of the discovery set is to predict antimicrobial gut compounds. In that sense, this activity is not annotated/known for the instances included in discovery sets, but are relevant to predict. That is, FDA-approved drugs are not annotated to have antimicrobial activity against the 40 microbes from gut used to train our best model, yet prediction of this activity may reveal a non-desired activity. The relevance of this prediction is dual: i) for known antimicrobials the predicted activity may reveal unspecificity, hence to reveal broad-spectrum antibiotics and ii) for compounds used to treat non-infectious diseases, to reveal possible antimicrobial activity. Thus, using the discovery set was not for comparison purposes, the training and testing sets were used for that goal. To clarify this, we state in the results section now: “We used the best model to predict NPCC with expected gut antimicrobial activity among FDA-approved drugs. The motivation to perform this prediction is not for testing purposes, as in the case of the training and testing sets used before. Hence, the set of compounds used in this prediction stage is referred to as the discovery set, because we aimed to discover potential compounds with gut antimicrobial activity”.

Reviewer 2 Report

The manuscript entitled “Heterologous machine learning for the identification of antimicrobial activity in human-targeted drugs” by Rodrigo A. Nava Lara with co-authors reports the development of novel statistical model to predict antimicrobial activity of chemical compounds.

To enhance the predictive power of the model, authors proposed to combine the experimental data available for both, small molecule compounds and peptides. Using different combinations of peptide and small compound datasets, a number of statistical models have been constructed and their predictive power was evaluated. Overall, the results demonstrate a feasibility to build reliable classifiers using the combined data available for both, small molecules and peptides. However, it remains unclear how the models reported in the manuscript are improved comparing to previously published data. As mentioned on page 9 lines 223-226 the performance of the best model “is comparable” with recently reported models.

One concern for the first part of the paper raised from the statement on page 2: “We propose that using molecular descriptors common to peptides and NPCC should increase the training set size and the reliability of the computational models”. To identify COMMON descriptors, two sets of the descriptors should be determined: one set for peptides and another one for compounds. Then, the descriptors identified for peptide set should be compared with descriptors identified for NPCC. The overlapping descriptors should be listed in a separate table. The model built with this set of overlapping descriptors should be provided and discussed. It would be helpful to discuss what descriptors contribute in classification of peptides and what particular descriptors are important for small molecule compounds.

The “Identifying broad-spectrum antibiotics among FDA-approved compounds” section of the paper is relatively weak and somehow confusing:

1.        It follows from Table 3 that there were a total of 133 compounds with antimicrobial activity. The model was able to correctly identify 72 of them. It indicates that the predictive power of the model is about 50%, and thus the model cannot be actually used to distinguish compound with and without antimicrobial activity.

2.       It is not indicated in the text that the training set was anyhow specifically divided into broad- and narrow-spectrum antibiotics. Thus, the model was not trained to classify these two groups, and cannot be applied for this particular task.

3.       It is indicated on page 7 that “We also added 73 NPCC that included 22 antifungal 168 compounds”. It is not clear how this set was generated, and why it was needed to add these compound to a set of the FDA approved drugs.

As a minor comment, all descriptors used to generate the models should be listed in a separate table, otherwise the data cannot be reproduced.

Author Response

1.     Overall, the results demonstrate a feasibility to build reliable classifiers using the combined data available for both, small molecules and peptides. However, it remains unclear how the models reported in the manuscript are improved comparing to previously published data. As mentioned on page 9 lines 223-226 the performance of the best model “is comparable” with recently reported models.

We appreciate the note; indeed now we realize that this concept is not clearly explained in the original text. We noted that it is not possible to directly compare the reliability of our best model with previous results, mainly because the model reported in our work was trained against 40 gut antimicrobial compounds reported last year (2018), and this is the first machine-learning report with that data. Previous works have been reported to classify antimicrobial compounds, but not necessarily against these 40 gut antimicrobial compounds. What we can compare though, is how learnable is our data set considering the reliability in classification. That is, we can compare how well the boundary between gut antimicrobial compounds and non-antimicrobial compounds was identified in our study with those from previous studies. For that aim, we compared the learning efficiency of our best model with a recent report that achieved the best classification of antimicrobial peptides from non-antimicrobial peptides. To clarify this, we now added the following sentence in the discussion section: “We can compare our best classifier with previous works in terms of the learnability of our classes, that is, how well gut antimicrobial compounds are differentiated from non-antimicrobial gut compounds. In that sense, the numeric performance achieved by the best classifier on the testing set (AUC=0.83) is comparable with the performance achieved with one of the best antimicrobial peptide classifiers (AUC=0.85) recently reported [24], indicating that the learnability of heterologous training sets is as good as those of only peptides”.

2.     One concern for the first part of the paper raised from the statement on page 2: “We propose that using molecular descriptors common to peptides and NPCC should increase the training set size and the reliability of the computational models”. To identify COMMON descriptors, two sets of the descriptors should be determined: one set for peptides and another one for compounds. Then, the descriptors identified for peptide set should be compared with descriptors identified for NPCC. The overlapping descriptors should be listed in a separate table. The model built with this set of overlapping descriptors should be provided and discussed. It would be helpful to discuss what descriptors contribute in classification of peptides and what particular descriptors are important for small molecule compounds.

This is an important point that we missed to clarify in the first version of our work. PadelDescriptor calculates 1444 molecular descriptors that are common to both peptides and NPCC; however, not all of these descriptors are useful for classification of gut antimicrobial compounds. Thus, the goal of our work is not to identify such common descriptors because all those are part of PadelDescriptor; instead, we aimed to identify molecular descriptors common to both peptides and NPCC useful to classify gut antimicrobial compounds. To clarify this, we state now in the introduction section: “The need to use common molecular descriptors between polypeptides and NPCC has been previously noted for protein-ligand recognition and protein folding, as a fundamental aspect to deal with induced-fit or conformer selection mechanisms for molecular recognition [19]; the aim of this work though, is not to find common descriptors to peptides and NPCC since there are already packages that solve this problem (see below). Here we propose that combining peptides and NPCC increases the training set size and this should improve the reliability of the computational models. The present work tests this proposal and validates the idea that heterelogous (NPCC and peptides) training sets render the best classifying models. We then show how this improved model may assist in the identification of broad-spectrum antibiotics on FDA-approved NPCC”.

3.     The “Identifying broad-spectrum antibiotics among FDA-approved compounds” section of the paper is relatively weak and somehow confusing: i) It follows from Table 3 that there were a total of 133 compounds with antimicrobial activity. The model was able to correctly identify 72 of them. It indicates that the predictive power of the model is about 50%, and thus the model cannot be actually used to distinguish compound with and without antimicrobial activity.

We appreciate this note to help us clarify this important aspect of our work. This was also a point noted by the other reviewer of our work and this helped us realize this limitation in describing correctly this aspect of our work. The discovery set cannot be used to test the reliability of the best classifier instead the training and testing sets are used for that purpose. The discovery set is used to predict gut antimicrobial compounds in a dataset that does not include any information about that activity. To clarify this aspect of the discovery set, we have now added in the results section: “We used the best model to predict NPCC with expected gut antimicrobial activity among FDA-approved drugs. The motivation to perform this prediction is not for testing purposes, as in the case of the training and testing sets used before. Hence, the set of compounds used in this prediction stage is referred to as the discovery set, because we aimed to discover potential compounds with gut antimicrobial activity”.

4.     The “Identifying broad-spectrum antibiotics among FDA-approved compounds” section of the paper is relatively weak and somehow confusing: ii) It is not indicated in the text that the training set was anyhow specifically divided into broad- and narrow-spectrum antibiotics. Thus, the model was not trained to classify these two groups, and cannot be applied for this particular task.

We appreciate this note. It is clear for us now we did not explain clearly the meaning of the discovery set on one side, and the way in which broad-spectrum antibiotics were defined. To clarify this, we also now state in the results section: “We used 756 FDA-approved compounds included in the ZINC database that were not part of the training or testing sets; these compounds included 111 antimicrobials and 645 compounds without any known antimicrobial activity; we also added 73 NPCC that included 22 antifungal compounds and 51 without any reported antifungal activity (see supplementary Table S22). We have previously reported that these 22 antifungals work through a mechanism (alter calcium intake [21]) different from antibacterial compounds (e.g., penicillin derivates, sulphonamides, etc), thus we expected our model to predict few of these compounds as antibacterials. FDA-approved compounds on the other hand are expected not to have, or to have minor, gut antimicrobial activity otherwise their secondary gastrointestinal effects would be significant. We would expect that FDA-approved drugs would be less likely predicted to act against non-pathogenic gut microbes than antifungals. To evaluate the reliability of our predictions using the discovery set, we considered that antibiotic compounds against the non-pathogenic gut flora among the FDA-approved drugs should be considered broad-spectrum antibiotics; please note that our classifier was not trained to predict this class of antibiotics, yet the combination of the predictions of our classifier on the FDA-approved drugs would render this information. The definition of broad-spectrum antibiotics is somehow arbitrary, for instance, it is considered that antibiotics that act on G(+) and G(-) are broad-spectrum antibiotics for some authors, while those acting against pathogenic and non-pathogenic microorganisms are classified as broad-spectrum antibiotics by others [22,23]. The list of broad-spectrum antibiotics was obtained from five recent works (see Methods), including 19 broad-spectrum and 3 narrow-spectrum antibiotics (see supplementary Table S22). We were able to identify 72 true positives (FDA-approved antibiotics against pathogenic microbes predicted to act against non-pathogenic gut microbes) in the discovery set that we predicted should be considered as broad-spectrum antibiotics (see Table 3)”. Tables 3 and 4 were also re-annotated to clarify this aspect.

Finally, we also clarify furthermore this aspect in the discussion section: “How significant is our finding that almost half of the FDA-approved antibiotics are predicted to have a broad-spectrum activity? To address this question, we included in the discovery set a group of antifungal compounds. All microorganisms used to train our models were bacteria, hence we expected that these antifungals that act through a mechanism different from those reported for bacteria would be unlikely predicted to act against bacteria; lets refer to this negative prediction as expectation-antifungal. On the other hand, most FDA-approved antibiotics should unlikely present antibiotic activity against gut microbes, otherwise these would frequently have secondary gastrointestinal effects on patients; lets refer to this negative prediction as expectation-FDA. Then, to address the significance of our findings about broad-spectrum antibiotics requires evaluating expectation-antifungal and expectation-FDA; if FDA-approved drugs are less likely to act on gut microbes than antifungals then expectation-FDA < expectation-antifungal. Indeed, we observed that FDA-approved antibiotics are twice as much less likely to act against gut microbes than antifungals. Thus, our results indicate that even when FDA-approved antibiotics are safer (do not act against non-pathogenic resident gut bacteria) than our control group (antifungals), we identified some of these compounds that need to be re-assessed as potential promoters of resistance among microbes for their potential broad-spectrum activity”.

5.     The “Identifying broad-spectrum antibiotics among FDA-approved compounds” section of the paper is relatively weak and somehow confusing: iii) It is indicated on page 7 that “We also added 73 NPCC that included 22 antifungal 168 compounds”. It is not clear how this set was generated, and why it was needed to add these compound to a set of the FDA approved drugs.

The purpose of these 22 antifungal compounds is to serve as a negative control group, that is, our group previously characterized many of these antifungal compounds to affect calcium intake, a mechanism not commonly found among antibiotics (e.g., penicillin derivates, sulphas, etc.), and consequently we expected these antifungal not to share molecular descriptors common to antibiotics, particularly with those gut antibiotics used to train our model. This selection has the limitation that we do not for sure that these antifungals wont have any antibiotic activity against those gut microbes, but it was merely a control group that seemed reasonable. To clarify this, we now added in the results section: “we also added 73 NPCC that included 22 antifungal compounds and 51 without any reported antifungal activity (see supplementary Table S22). We have previously reported that these 22 antifungals work through a mechanism (alter calcium intake [21]) different from antibacterial compounds (e.g., penicillin derivates, sulphonamides, etc), thus we expected our model to predict few of these compounds as antibacterials. FDA-approved compounds on the other hand are expected not to have, or to have minor, gut antimicrobial activity otherwise their secondary gastrointestinal effects would be significant. We would expect that FDA-approved drugs would be less likely predicted to act against non-athogenic gut microbes than antifungals”.

6.     As a minor comment, all descriptors used to generate the models should be listed in a separate table, otherwise the data cannot be reproduced.

We originally included all training and testing sets (that included every descriptors used in each case) as supplemental tables, but we did indeed merged all training and testing sets of the same class (only peptides, only NPCC and heterologous) in a single file. We are now including all descriptors used in all training and testing sets in ARFF format in separate tables.

Round 2

Reviewer 1 Report

I consider that the manuscript can be published in the present form.

Reviewer 2 Report

The authors have addressed all major issues and concerns. I do not have any further comments.